# KRAB-zinc finger protein gene expansion in response to active retrotransposons in the murine lineage

Gernot Wolf[1†], Alberto de Iaco[2], Ming-An Sun[1], Melania Bruno[1], Matthew Tinkham[1], Don Hoang[1], Apratim Mitra[1], Sherry Ralls[1], Didier Trono[2], Todd S Macfarlan[1]*

[1]The Eunice Kennedy Shriver National Institute of Child Health and Human Development, The National Institutes of Health, Bethesda, United States; [2]School of Life Sciences, École Polytechnique Fédérale de Lausanne (EPFL), Lausanne, Switzerland

**Abstract** The Krüppel-associated box zinc finger protein (KRAB-ZFP) family diversified in mammals. The majority of human KRAB-ZFPs bind transposable elements (TEs), however, since most TEs are inactive in humans it is unclear whether KRAB-ZFPs emerged to suppress TEs. We demonstrate that many recently emerged murine KRAB-ZFPs also bind to TEs, including the active ETn, IAP, and L1 families. Using a CRISPR/Cas9-based engineering approach, we genetically deleted five large clusters of KRAB-ZFPs and demonstrate that target TEs are de-repressed, unleashing TE-encoded enhancers. Homozygous knockout mice lacking one of two KRAB-ZFP gene clusters on chromosome 2 and chromosome 4 were nonetheless viable. In pedigrees of chromosome 4 cluster KRAB-ZFP mutants, we identified numerous novel ETn insertions with a modest increase in mutants. Our data strongly support the current model that recent waves of retrotransposon activity drove the expansion of KRAB-ZFP genes in mice and that many KRAB-ZFPs play a redundant role restricting TE activity.

*For correspondence:
todd.macfarlan@nih.gov

Present address: †CeMM Research Center for Molecular Medicine of the Austrian Academy of Sciences, Vienna, Austria

Competing interests: The authors declare that no competing interests exist.

## Introduction

Nearly half of the human and mouse genomes consist of transposable elements (TEs). TEs shape the evolution of species, serving as a source for genetic innovation (*Chuong et al., 2016*; *Frank and Feschotte, 2017*). However, TEs also potentially harm their hosts by insertional mutagenesis, gene deregulation and activation of innate immunity (*Maksakova et al., 2006*; *Kano et al., 2007*; *Brodziak et al., 2012*; *Hancks and Kazazian, 2016*). To protect themselves from TE activity, host organisms have developed a wide range of defense mechanisms targeting virtually all steps of the TE life cycle (*Dewannieux and Heidmann, 2013*). In tetrapods, KRAB zinc finger protein (KRAB-ZFP) genes have amplified and diversified, likely in response to TE colonization (*Thomas and Schneider, 2011*; *Najafabadi et al., 2015*; *Wolf et al., 2015a*; *Wolf et al., 2015b*; *Imbeault et al., 2017*). Conventional ZFPs bind DNA using tandem arrays of C2H2 zinc finger domains, each capable of specifically interacting with three nucleotides, whereas some zinc fingers can bind two or four nucleotides and include DNA backbone interactions depending on target DNA structure (*Patel et al., 2018*). This allows KRAB-ZFPs to flexibly bind to large stretches of DNA with high affinity. The KRAB domain binds the corepressor KAP1, which in turn recruits histone modifying enzymes including the NuRD histone deacetylase complex and the H3K9-specific methylase SETDB1 (*Schultz et al., 2002*; *Sripathy et al., 2006*), which induces persistent and heritable gene silencing (*Groner et al., 2010*). Deletion of KAP1 (*Rowe et al., 2010*) or SETDB1 (*Matsui et al., 2010*) in mouse embryonic stem (ES) cells induces TE reactivation and cell death, but only minor phenotypes in differentiated cells,

suggesting KRAB-ZFPs are most important during early embryogenesis where they mark TEs for stable epigenetic silencing that persists through development. However, SETDB1-containing complexes are also required to repress TEs in primordial germ cells (*Liu et al., 2014*) and adult tissues (*Ecco et al., 2016*), indicating KRAB-ZFPs are active beyond early development.

TEs, especially long terminal repeat (LTR) retrotransposons, also known as endogenous retroviruses (ERVs), can affect expression of neighboring genes through their promoter and enhancer functions (*Macfarlan et al., 2012*; *Wang et al., 2014*; *Thompson et al., 2016*). KAP1 deletion in mouse ES cells causes rapid gene deregulation (*Rowe et al., 2013*), indicating that KRAB-ZFPs may regulate gene expression by recruiting KAP1 to TEs. Indeed, *Zfp809* knock-out (KO) in mice resulted in transcriptional activation of a handful of genes in various tissues adjacent to ZFP809-targeted VL30-Pro elements (*Wolf et al., 2015b*). It has therefore been speculated that KRAB-ZFPs bind to TE sequences to domesticate them for gene regulatory innovation (*Ecco et al., 2017*). This idea is supported by the observation that many human KRAB-ZFPs target TE groups that have lost their coding potential millions of years ago and that KRAB-ZFP target sequences within TEs are in some cases under purifying selection (*Imbeault et al., 2017*). However, there are also clear signs of an evolutionary arms-race between human TEs and KRAB-ZFPs (*Jacobs et al., 2014*), indicating that some KRAB-ZFPs may limit TE mobility for stretches of evolutionary time, prior to their ultimate loss from the genome or adaptation for other regulatory functions. Here we use the laboratory mouse, which has undergone a recent expansion of the KRAB-ZFP family, to determine the in vivo requirement of the majority of evolutionarily young KRAB-ZFP genes.

## Results

### Mouse KRAB-ZFPs target retrotransposons

We analyzed the RNA expression profiles of mouse KRAB-ZFPs across a wide range of tissues to identify candidates active in early embryos/ES cells. While the majority of KRAB-ZFPs are expressed at low levels and uniformly across tissues, a group of KRAB-ZFPs are highly and almost exclusively expressed in ES cells (*Figure 1—figure supplement 1A*). About two thirds of these KRAB-ZFPs are physically linked in two clusters on chromosome 2 (Chr2-cl) and 4 (Chr4-cl) (*Figure 1—figure supplement 1B*). These two clusters encode 40 and 21 KRAB-ZFP annotated genes, respectively, which, with one exception on Chr4-cl, do not have orthologues in rat or any other sequenced mammals (*Supplementary file 1*). The KRAB-ZFPs within these two genomic clusters also group together phylogenetically (*Figure 1—figure supplement 1C*), indicating these gene clusters arose by a series of recent segmental gene duplications (*Kauzlaric et al., 2017*).

To determine the binding sites of the KRAB-ZFPs within these and other gene clusters, we expressed epitope-tagged KRAB-ZFPs using stably integrating vectors in mouse embryonic carcinoma (EC) or ES cells (*Table 1*, *Supplementary file 1*) and performed chromatin immunoprecipitation followed by deep sequencing (ChIP-seq). We then determined whether the identified binding sites are significantly enriched over annotated TEs and used the non-repetitive peak fraction to identify binding motifs. We discarded 7 of 68 ChIP-seq datasets because we could not obtain a binding motif or a target TE and manual inspection confirmed low signal to noise ratio. Of the remaining 61 KRAB-ZFPs, 51 significantly overlapped at least one TE subfamily (*adjusted p-value*<1e-5). Altogether, 81 LTR retrotransposon, 18 LINE, 10 SINE and one DNA transposon subfamilies were targeted by at least one of the 51 KRAB-ZFPs (*Figure 1A* and *Supplementary file 1*). Chr2-cl KRAB-ZFPs preferably bound IAPEz retrotransposons and L1-type LINEs, while Chr4-cl KRAB-ZFPs targeted various retrotransposons, including the closely related MMETn (hereafter referred to as ETn) and ETnERV (also known as MusD) elements (*Figure 1A*). ETn elements are non-autonomous LTR retrotransposons that require trans-complementation by the fully coding ETnERV elements that contain Gag, Pro and Pol genes (*Ribet et al., 2004*). These elements have accumulated to ~240 and~100 copies in the reference C57BL/6 genome, respectively, with ~550 solitary LTRs (*Baust et al., 2003*). Both ETn and ETnERVs are still active, generating polymorphisms and mutations in several mouse strains (*Gagnier et al., 2019*). The validity of our ChIP-seq screen was confirmed by the identification of binding motifs - which often resembled the computationally predicted motifs (*Figure 1—figure supplement 2A*) - for the majority of screened KRAB-ZFPs (*Supplementary file 1*). Moreover, predicted and experimentally determined motifs were found in

**Table 1.** KRAB-ZFP genes clusters in the mouse genome that were investigated in this study.
* Number of protein-coding KRAB-ZFP genes identified in a previously published screen (*Imbeault et al., 2017*) and the ChIP-seq data column indicates the number of KRAB-ZFPs for which ChIP-seq was performed in this study.

| Cluster | Location | Size (Mb) | # of KRAB-ZFPs* | ChIP-seq data |
|---------|----------|-----------|-----------------|---------------|
| Chr2 | Chr2 qH4 | 3.1 | 40 | 17 |
| Chr4 | Chr4 qE1 | 2.3 | 21 | 19 |
| Chr10 | Chr10 qC1 | 0.6 | 6 | 1 |
| Chr13.1 | Chr13 qB3 | 1.2 | 6 | 2 |
| Chr13.2 | Chr13 qB3 | 0.8 | 26 | 12 |
| Chr8 | Chr8 qB3.3 | 0.1 | 4 | 4 |
| Chr9 | Chr9 qA3 | 0.1 | 4 | 2 |
| Other | - | - | 248 | 4 |

targeted TEs in most cases (*Supplementary file 1*), and reporter repression assays confirmed KRAB-ZFP induced silencing for all the tested sequences (*Figure 1—figure supplement 2B*). Finally, we observed KAP1 and H3K9me3 enrichment at most of the targeted TEs in wild type ES cells, indicating that most of these KRAB-ZFPs are functionally active in the early embryo (*Figure 1A*).

We generally observed that KRAB-ZFPs present exclusively in mouse target TEs that are restricted to the mouse genome, indicating KRAB-ZFPs and their targets emerged together. For example, several mouse-specific KRAB-ZFPs in Chr2-cl and Chr4-cl target IAP and ETn elements which are only found in the mouse genome and are highly active. This is the strongest data to date supporting that recent KRAB-ZFP expansions in these young clusters is a response to recent TE activity. Likewise, ZFP599 and ZFP617, both conserved in Muroidea, bind to various ORR1-type LTRs which are present in the rat genome (*Supplementary file 1*). However, ZFP961, a KRAB-ZFP encoded on a small gene cluster on chromosome 8 that is conserved in Muroidea targets TEs that are only found in the mouse genome (e.g. ETn), a paradox we have previously observed with ZFP809, which also targets TEs that are evolutionarily younger than itself (*Wolf et al., 2015b*). The ZFP961 binding site is located at the 5' end of the internal region of ETn and ETnERV elements, a sequence that usually contains the primer binding site (PBS), which is required to prime retroviral reverse transcription. Indeed, the ZFP961 motif closely resembles the PBS[Lys1,2] (*Figure 1—figure supplement 3A*), which had been previously identified as a KAP1-dependent target of retroviral repression (*Yamauchi et al., 1995*; *Wolf et al., 2008*). Repression of the PBS[Lys1,2] by ZFP961 was also confirmed in reporter assays (*Figure 1—figure supplement 2B*), indicating that ZFP961 is likely responsible for this silencing effect.

To further test the hypothesis that KRAB-ZFPs target sites necessary for retrotransposition, we utilized previously generated ETn and ETnERV retrotransposition reporters in which we mutated KRAB-ZFP binding sites (*Ribet et al., 2004*). Whereas the ETnERV reporters are sufficient for retrotransposition, the ETn reporter requires ETnERV genes supplied in trans. We tested and confirmed that the REX2/ZFP600 and GM13051 binding sites within these TEs are required for efficient retrotransposition (*Figure 1—figure supplement 3B*). REX2 and ZFP600 both bind a target about 200 bp from the start of the internal region (*Figure 1B*), a region that often encodes the packaging signal. GM13051 binds a target coding for part of a highly structured mRNA export signal (*Legiewicz et al., 2010*) near the 3' end of the internal region of ETn (*Figure 1—figure supplement 3C*). Both signals are characterized by stem-loop intramolecular base-pairing in which a single mutation can disrupt loop formation. This indicates that at least some KRAB-ZFPs evolved to bind functionally essential target sequences which cannot easily evade repression by mutation.

Our KRAB-ZFP ChIP-seq dataset also provided unique insights into the emergence of new KRAB-ZFPs and binding patterns. The Chr4-cl KRAB-ZFPs REX2 and ZFP600 bind to the same target within ETn but with varying affinity (*Figure 1C*). Comparison of the amino acids responsible for DNA contact revealed a high similarity between REX2 and ZFP600, with the main differences at the most C-terminal zinc fingers. Additionally, we found that GM30910, another KRAB-ZFP encoded in the

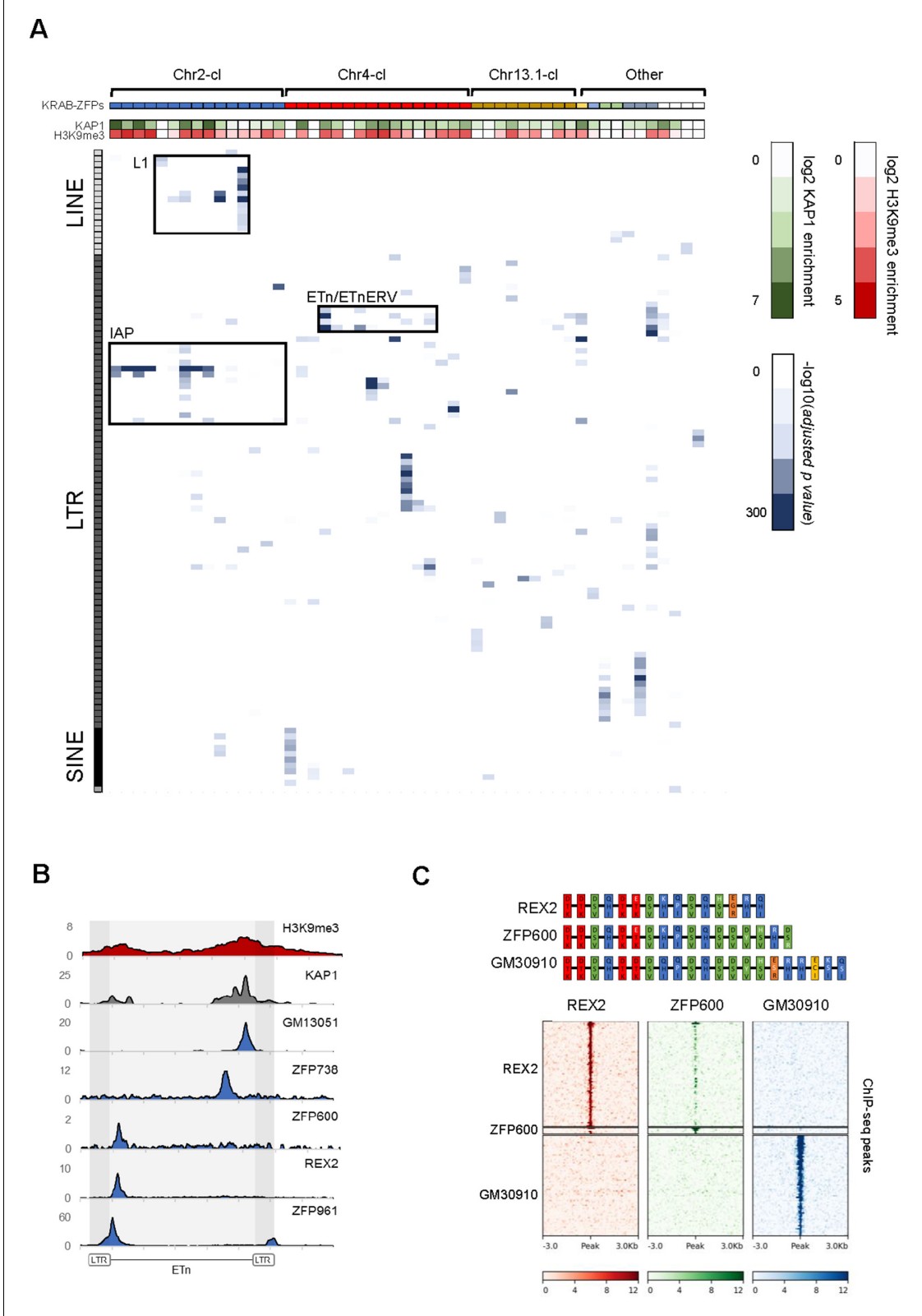

**Figure 1.** Genome-wide binding patterns of mouse KRAB-ZFPs. (**A**) Probability heatmap of KRAB-ZFP binding to TEs. Blue color intensity (main field) corresponds to -log10 (*adjusted p-value*) enrichment of ChIP-seq peak overlap with TE groups (Fisher's exact test). The green/red color intensity (top panel) represents mean KAP1 (GEO accession: GSM1406445) and H3K9me3 (GEO accession: GSM1327148) enrichment (respectively) at peaks overlapping significantly targeted TEs (*adjusted p-value*<1e-5) in WT ES cells. (**B**) Summarized ChIP-seq signal for indicated KRAB-ZFPs and previously

*Figure 1 continued on next page*

*Figure 1 continued*

published KAP1 and H3K9me3 in WT ES cells across 127 intact ETn elements. (**C**) Heatmaps of KRAB-ZFP ChIP-seq signal at ChIP-seq peaks. For better comparison, peaks for all three KRAB-ZFPs were called with the same parameters (p<1e-10, peak enrichment >20). The top panel shows a schematic of the arrangement of the contact amino acid composition of each zinc finger. Zinc fingers are grouped and colored according to similarity, with amino acid differences relative to the five consensus fingers highlighted in white.

The online version of this article includes the following source data and figure supplement(s) for figure 1:

**Source data 1.** KRAB-ZFP expression in 40 mouse tissues and cell lines (ENCODE).
**Source data 2.** Probability heatmap of KRAB-ZFP binding to TEs.
**Figure supplement 1.** ES cell-specific expression of KRAB-ZFP gene clusters.
**Figure supplement 2.** KRAB-ZFP binding motifs and their repression activity.
**Figure supplement 3.** KRAB-ZFP binding to ETn retrotransposons.

Chr4-cl, also shows a strong similarity to both KRAB-ZFPs yet targets entirely different groups of TEs (*Figure 1C* and *Supplementary file 1*). Together with previously shown data (*Ecco et al., 2016*), this example highlights how addition of a few new zinc fingers to an existing array can entirely shift the mode of DNA binding.

## Genetic deletion of KRAB-ZFP gene clusters leads to retrotransposon reactivation

The majority of KRAB-ZFP genes are harbored in large, highly repetitive clusters that have formed by successive complex segmental duplications (*Kauzlaric et al., 2017*), rendering them inaccessible to conventional gene targeting. We therefore developed a strategy to delete entire KRAB-ZFP gene clusters in ES cells (including the Chr2-cl and Chr4-cl as well as two clusters on chromosome 13 and a cluster on chromosome 10) using two CRISPR/Cas9 gRNAs targeting unique regions flanking each cluster, and short single-stranded repair oligos with homologies to both sides of the projected cut sites. Using this approach, we generated five cluster KO ES cell lines in at least two biological replicates and performed RNA sequencing (RNA-seq) to determine TE expression levels. Strikingly, four of the five cluster KO ES cells exhibited distinct TE reactivation phenotypes (*Figure 2A*). Chr2-cl KO resulted in reactivation of several L1 subfamilies as well as RLTR10 (up to more than 100-fold as compared to WT) and IAPEz ERVs. In contrast, the most strongly upregulated TEs in Chr4-cl KO cells were ETn/ETnERV (up to 10-fold as compared to WT), with several other ERV groups modestly reactivated. ETn/ETnERV elements were also upregulated in Chr13.2-cl KO ES cells while the only upregulated ERVs in Chr13.1-cl KO ES cells were MMERVK10C elements (*Figure 2A*). Most reactivated retrotransposons were targeted by at least one KRAB-ZFP that was encoded in the deleted cluster (*Figure 2A* and *Supplementary file 1*), indicating a direct effect of these KRAB-ZFPs on TE expression levels. Furthermore, we observed a loss of KAP1 binding and H3K9me3 at several TE subfamilies that are targeted by at least one KRAB-ZFP within the deleted Chr2-cl and Chr4-cl (*Figure 2B*, *Figure 2—figure supplement 1A*), including L1, ETn and IAPEz elements. Using reduced representation bisulfite sequencing (RRBS-seq), we found that a subset of KRAB-ZFP bound TEs were partially hypomethylated in Chr4-cl KO ES cells, but only when grown in genome-wide hypomethylation-inducing conditions (*Blaschke et al., 2013*; *Figure 2C* and *Supplementary file 2*). These data are consistent with the hypothesis that KRAB-ZFPs/KAP1 are not required to establish DNA methylation, but under certain conditions they protect specific TEs and imprint control regions from genome-wide demethylation (*Leung et al., 2014*; *Deniz et al., 2018*).

## KRAB-ZFP cluster deletions license TE-borne enhancers

We next used our RNA-seq datasets to determine the effect of KRAB-ZFP cluster deletions on gene expression. We identified 195 significantly upregulated and 130 downregulated genes in Chr4-cl KO ES cells, and 108 upregulated and 59 downregulated genes in Chr2-cl KO ES cells (excluding genes on the deleted cluster) (*Figure 3A*). To address whether gene deregulation in Chr2-cl and Chr4-cl KO ES cells is caused by nearby TE reactivation, we determined whether genes near certain TE subfamilies are more frequently deregulated than random genes. We found a strong correlation of gene upregulation and TE proximity for several TE subfamilies, of which many became transcriptionally activated themselves (*Figure 3B*). For example, nearly 10% of genes that are located within 100 kb (up- or downstream of the TSS) of an ETn element are upregulated in Chr4-cl KO ES cells, as

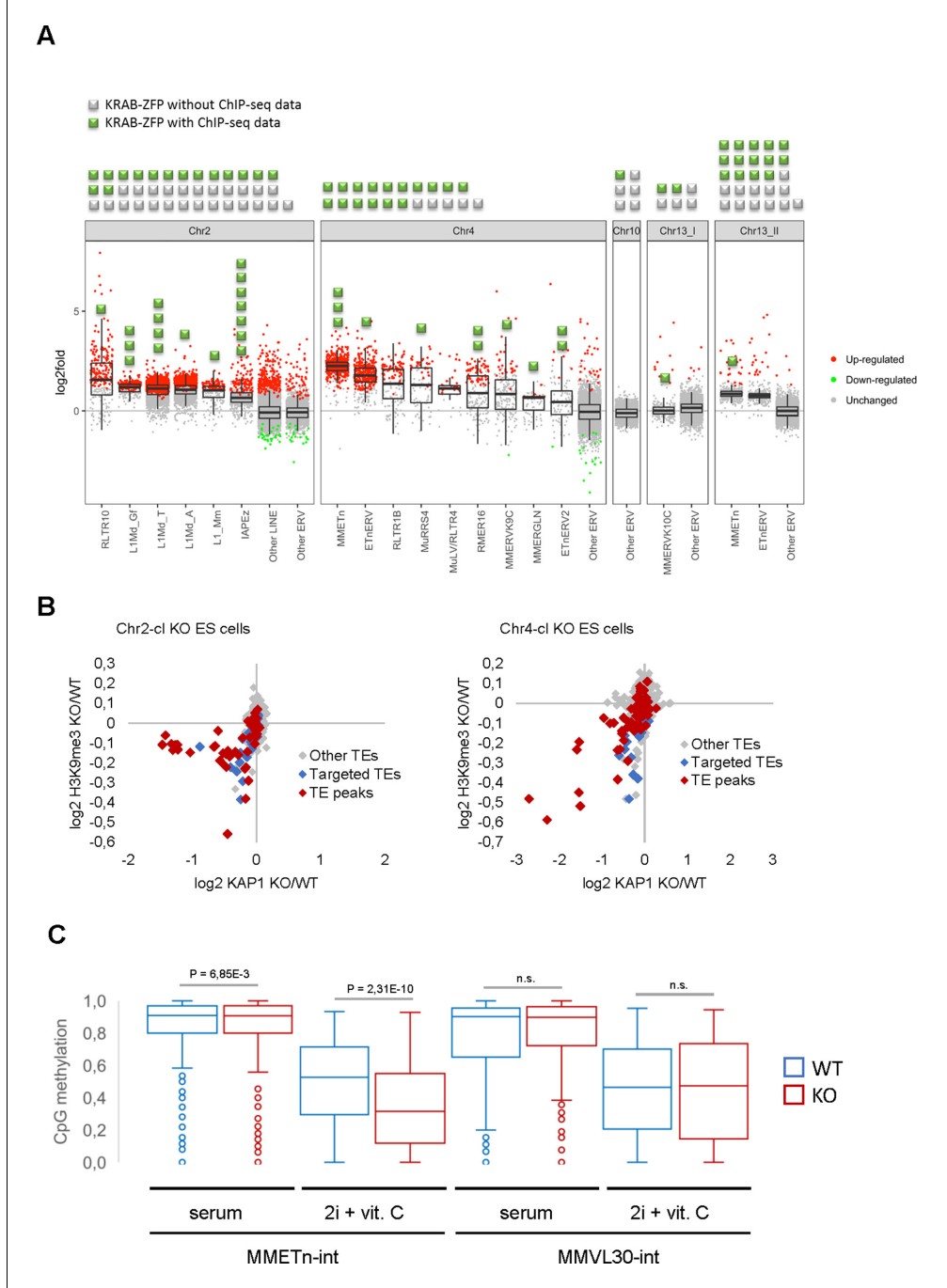

**Figure 2.** Retrotransposon reactivation in KRAB-ZFP cluster KO ES cells. (**A**) RNA-seq analysis of TE expression in five KRAB-ZFP cluster KO ES cells. Green and grey squares on top of the panel represent KRAB-ZFPs with or without ChIP-seq data, respectively, within each deleted gene cluster. Reactivated TEs that are bound by one or several KRAB-ZFPs are indicated by green squares in the panel. Significantly up- and downregulated elements (*adjusted p-value*<0.05) are highlighted in red and green, respectively. (**B**) Differential KAP1 binding and H3K9me3 enrichment at TE groups (summarized across all insertions) in Chr2-cl and Chr4-cl KO ES cells. TE groups targeted by one or several KRAB-ZFPs encoded within the deleted clusters are highlighted in blue (differential enrichment over the entire TE sequences) and red (differential enrichment at TE regions that overlap with KRAB-ZFP ChIP-seq peaks). (**C**) DNA methylation status of CpG sites at indicated TE groups in WT and Chr4-cl KO ES cells grown in serum containing media or in hypomethylation-inducing media (2i + Vitamin C). P-values were calculated using paired t-test.

The online version of this article includes the following source data and figure supplement(s) for figure 2:

**Source data 1.** Differential H3K9me3 and KAP1 distribution in WT and KRAB-ZFP cluster KO ES cells at TE families and KRAB-ZFP bound TE insertions.
**Figure supplement 1.** Epigenetic changes at TEs and TE-borne enhancers in KRAB-ZFP cluster KO ES cells.

compared to 0.8% of all genes. In Chr2-cl KO ES cells, upregulated genes were significantly enriched near various LINE groups but also IAPEz-int and RLTR10-int elements, indicating that TE-binding KRAB-ZFPs in these clusters limit the potential activating effects of TEs on nearby genes.

While we generally observed that TE-associated gene reactivation is not caused by elongated or spliced transcription starting at the retrotransposons, we did observe that the strength of the effect of ETn elements on gene expression is stronger on genes in closer proximity. About 25% of genes located within 20 kb of an ETn element, but only 5% of genes located at a distance between 50 and 100 kb from the nearest ETn insertion, become upregulated in Chr4-cl KO ES cells. Importantly however, the correlation is still significant for genes that are located at distances between 50 and 100 kb from the nearest ETn insertion, indicating that ETn elements can act as long-range enhancers of gene expression in the absence of KRAB-ZFPs that target them. To confirm that Chr4-cl KRAB-ZFPs such as GM13051 block ETn-borne enhancers, we tested the ability of a putative ETn enhancer to activate transcription in a reporter assay. For this purpose, we cloned a 5 kb fragment spanning from the GM13051 binding site within the internal region of a truncated ETn insertion to the first exon of the *Cd59a* gene, which is strongly activated in Chr4-cl KO ES cells (*Figure 2—figure supplement 1B*). We observed strong transcriptional activity of this fragment which was significantly higher in Chr4-cl KO ES cells. Surprisingly, this activity was reduced to background when the internal segment of the ETn element was not included in the fragment, suggesting the internal segment of the ETn element, but not its LTR, contains a Chr4-cl KRAB-ZFP sensitive enhancer. To further corroborate these findings, we genetically deleted an ETn element that is located about 60 kb from the TSS of *Chst1*, one of the top-upregulated genes in Chr4-cl KO ES cells (*Figure 3C*). RT-qPCR analysis revealed that the *Chst1* upregulation phenotype in Chr4-cl KO ES cells diminishes when the ETn insertion is absent, providing direct evidence that a KRAB-ZFP controlled ETn-borne enhancer regulates *Chst1* expression (*Figure 3D*). Furthermore, ChIP-seq confirmed a general increase of H3K4me3, H3K4me1 and H3K27ac marks at ETn elements in Chr4-cl KO ES cells (*Figure 3E*). Notably, enhancer marks were most pronounced around the GM13051 binding site near the 3' end of the internal region, confirming that the enhancer activity of ETn is located on the internal region and not on the LTR.

## ETn retrotransposition in Chr4-cl KO and WT mice

IAP, ETn/ETnERV and MuLV/RLTR4 retrotransposons are highly polymorphic in inbred mouse strains (*Nellåker et al., 2012*), indicating that these elements are able to mobilize in the germ line. Since these retrotransposons are upregulated in Chr2-cl and Chr4-cl KO ES cells, we speculated that these KRAB-ZFP clusters evolved to minimize the risks of insertional mutagenesis by retrotransposition. To test this, we generated Chr2-cl and Chr4-cl KO mice via ES cell injection into blastocysts, and after germ line transmission we genotyped the offspring of heterozygous breeding pairs. While the offspring of Chr4-cl KO/WT parents were born close to Mendelian ratios in pure C57BL/6 and mixed C57BL/6 129Sv matings, one Chr4-cl KO/WT breeding pair gave birth to significantly fewer KO mice than expected (p-value=0.022) (*Figure 4—figure supplement 1A*). Likewise, two out of four Chr2-cl KO breeding pairs on mixed C57BL/6 129Sv matings failed to give birth to a single KO offspring (p-value<0.01) while the two other mating pairs produced KO offspring at near Mendelian ratios (*Figure 4—figure supplement 1A*). Altogether, these data indicate that KRAB-ZFP clusters are not absolutely essential in mice, but that genetic and/or epigenetic factors may contribute to reduced viability.

We reasoned that retrotransposon activation could account for the reduced viability of Chr2-cl and Chr4-cl KO mice in some matings. However, since only rare matings produced non-viable KO embryos, we instead turned to the viable KO mice to assay for increased transposon activity. RNA-seq in blood, brain and testis revealed that, with a few exceptions, retrotransposons upregulated in Chr2 and Chr4 KRAB-ZFP cluster KO ES cells are not expressed at higher levels in adult tissues (*Figure 4—figure supplement 1B*). Likewise, no strong transcriptional TE reactivation phenotype was observed in liver and kidney of Chr4-cl KO mice (data not shown) and ChIP-seq with antibodies against H3K4me1, H3K4me3 and H3K27ac in testis of Chr4-cl WT and KO mice revealed no increase of active histone marks at ETn elements or other TEs (data not shown). This indicates that Chr2-cl and Chr4-cl KRAB-ZFPs are primarily required for TE repression during early development. This is consistent with the high expression of these KRAB-ZFPs uniquely in ES cells (*Figure 1—figure supplement 1A*). To determine whether retrotransposition occurs at a higher frequency in Chr4-cl KO

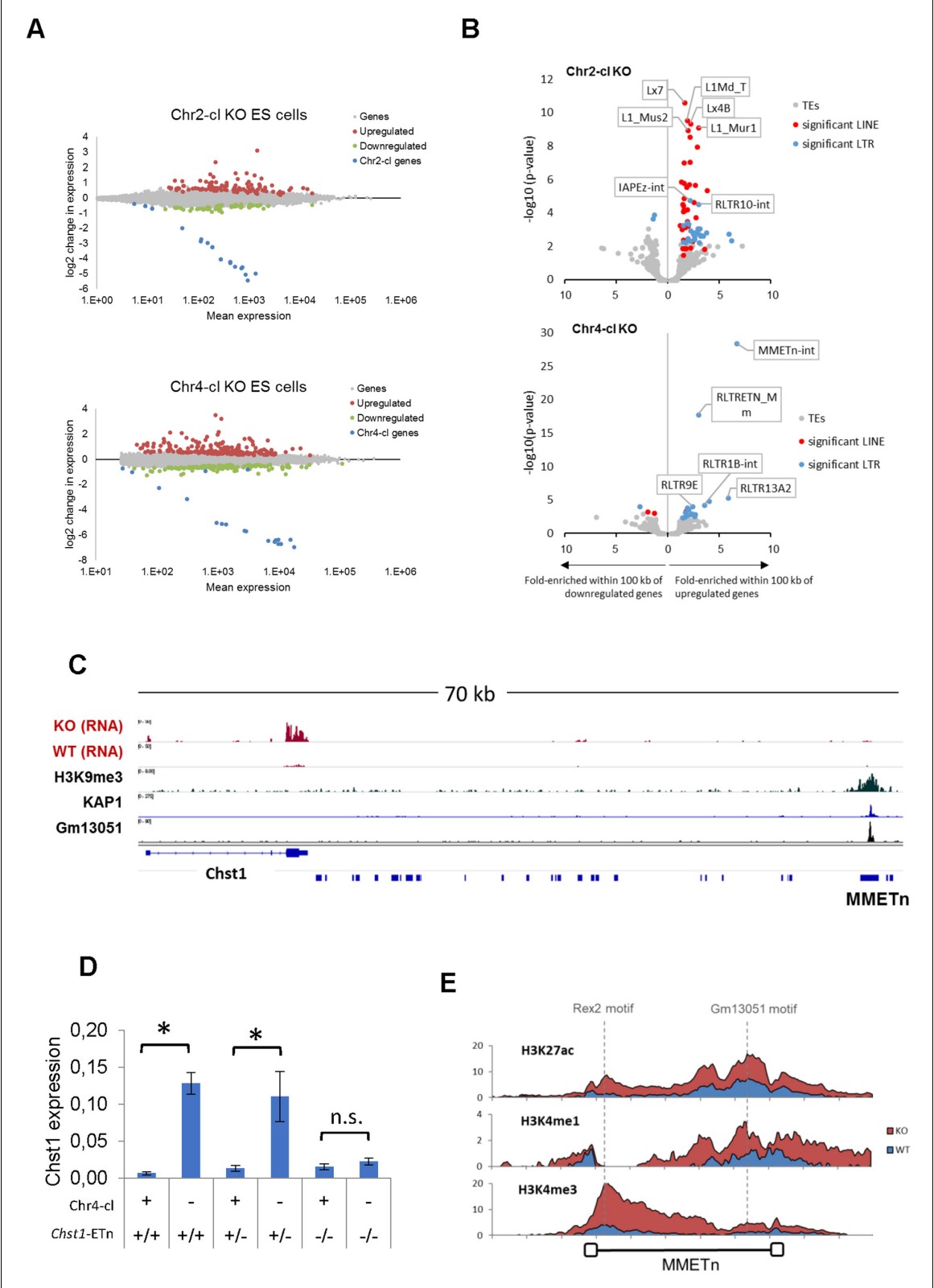

**Figure 3.** TE-dependent gene activation in KRAB-ZFP cluster KO ES cells. (**A**) Differential gene expression in Chr2-cl and Chr4-cl KO ES cells. Significantly up- and downregulated genes (*adjusted p-value*<0.05) are highlighted in red and green, respectively, KRAB-ZFP genes within the deleted clusters are shown in blue. (**B**) Correlation of TEs and gene deregulation. Plots show enrichment of TE groups within 100 kb of up- and downregulated genes relative to all genes. Significantly overrepresented LTR and LINE groups (*adjusted p-value*<0.1) are highlighted in blue and red, respectively. (**C**)
*Figure 3 continued on next page*

Figure 3 continued

Schematic view of the downstream region of *Chst1* where a 5' truncated ETn insertion is located. ChIP-seq (Input subtracted from ChIP) data for overexpressed epitope-tagged Gm13051 (a Chr4-cl KRAB-ZFP) in F9 EC cells, and re-mapped KAP1 (GEO accession: GSM1406445) and H3K9me3 (GEO accession: GSM1327148) in WT ES cells are shown together with RNA-seq data from Chr4-cl WT and KO ES cells (mapped using Bowtie (-a -m 1 —strata -v 2) to exclude reads that cannot be uniquely mapped). (D) RT-qPCR analysis of Chst1 mRNA expression in Chr4-cl WT and KO ES cells with or without the CRISPR/Cas9 deleted ETn insertion near *Chst1*. Values represent mean expression (normalized to Gapdh) from three biological replicates per sample (each performed in three technical replicates) in arbitrary units. Error bars represent standard deviation and asterisks indicate significance (p<0.01, Student's t-test). n.s.: not significant. (E) Mean coverage of ChIP-seq data (Input subtracted from ChIP) in Chr4-cl WT and KO ES cells over 127 full-length ETn insertions. The binding sites of the Chr4-cl KRAB-ZFPs Rex2 and Gm13051 are indicated by dashed lines.

mice during development, we screened for novel ETn (ETn/ETnERV) and MuLV (MuLV/RLTR4_MM) insertions in viable Chr4-cl KO mice. For this purpose, we developed a capture-sequencing approach to enrich for ETn/MuLV DNA and flanking sequences from genomic DNA using probes that hybridize with the 5' and 3' ends of ETn and MuLV LTRs prior to deep sequencing. We screened genomic DNA samples from a total of 76 mice, including 54 mice from ancestry-controlled Chr4-cl KO matings in various strain backgrounds, the two ES cell lines the Chr4-cl KO mice were generated from, and eight mice from a Chr2-cl KO mating which served as a control (since ETn and MuLVs are not activated in Chr2-cl KO ES cells) (*Supplementary file 4*). Using this approach, we were able to enrich reads mapping to ETn/MuLV LTRs about 2,000-fold compared to genome sequencing without capture. ETn/MuLV insertions were determined by counting uniquely mapped reads that were paired with reads mapping to ETn/MuLV elements (see materials and methods for details). To assess the efficiency of the capture approach, we determined what proportion of a set of 309 largely intact (two LTRs flanking an internal sequence) reference ETn elements could be identified using our sequencing data. 95% of these insertions were called with high confidence in the majority of our samples (data not shown), indicating that we are able to identify ETn insertions at a high recovery rate.

Using this dataset, we first confirmed the polymorphic nature of both ETn and MuLV retrotransposons in laboratory mouse strains (*Figure 4—figure supplement 2A*), highlighting the potential of these elements to retrotranspose. To identify novel insertions, we filtered out insertions that were supported by ETn/MuLV-paired reads in more than one animal. While none of the 54 ancestry-controlled mice showed a single novel MuLV insertion, we observed greatly varying numbers of up to 80 novel ETn insertions in our pedigree (*Figure 4A*).

To validate some of the novel ETn insertions, we designed specific PCR primers for five of the insertions and screened genomic DNA of the mice in which they were identified as well as their parents. For all tested insertions, we were able to amplify their flanking sequence and show that these insertions are absent in their parents (*Figure 4—figure supplement 3A*). To confirm their identity, we amplified and sequenced three of the novel full-length ETn insertions. Two of these elements (Genbank accession: MH449667-68) resembled typical ETnII elements with identical 5' and 3' LTRs and target site duplications (TSD) of 4 or 6 bp, respectively. The third sequenced element (MH449669) represented a hybrid element that contains both ETnI and MusD (ETnERV) sequences. Similar insertions can be found in the B6 reference genome; however, the identified novel insertion has a 2.5 kb deletion of the 5' end of the internal region. Additionally, the 5' and 3' LTR of this element differ in one nucleotide near the start site and contain an unusually large 248 bp TSD (containing a SINE repeat) indicating that an improper integration process might have truncated this element.

Besides novel ETn insertions that were only identified in one specific animal, we also observed three ETn insertions that could be detected in several siblings but not in their parents or any of the other screened mice. This strongly indicates that these retrotransposition events occurred in the germ line of the parents from which they were passed on to some of their offspring. One of these germ line insertions was evidently passed on from the offspring to the next generation (*Figure 4A*). As expected, the read numbers supporting these novel germ line insertions were comparable to the read numbers that were found in the flanking regions of annotated B6 ETn insertions (*Figure 4—*

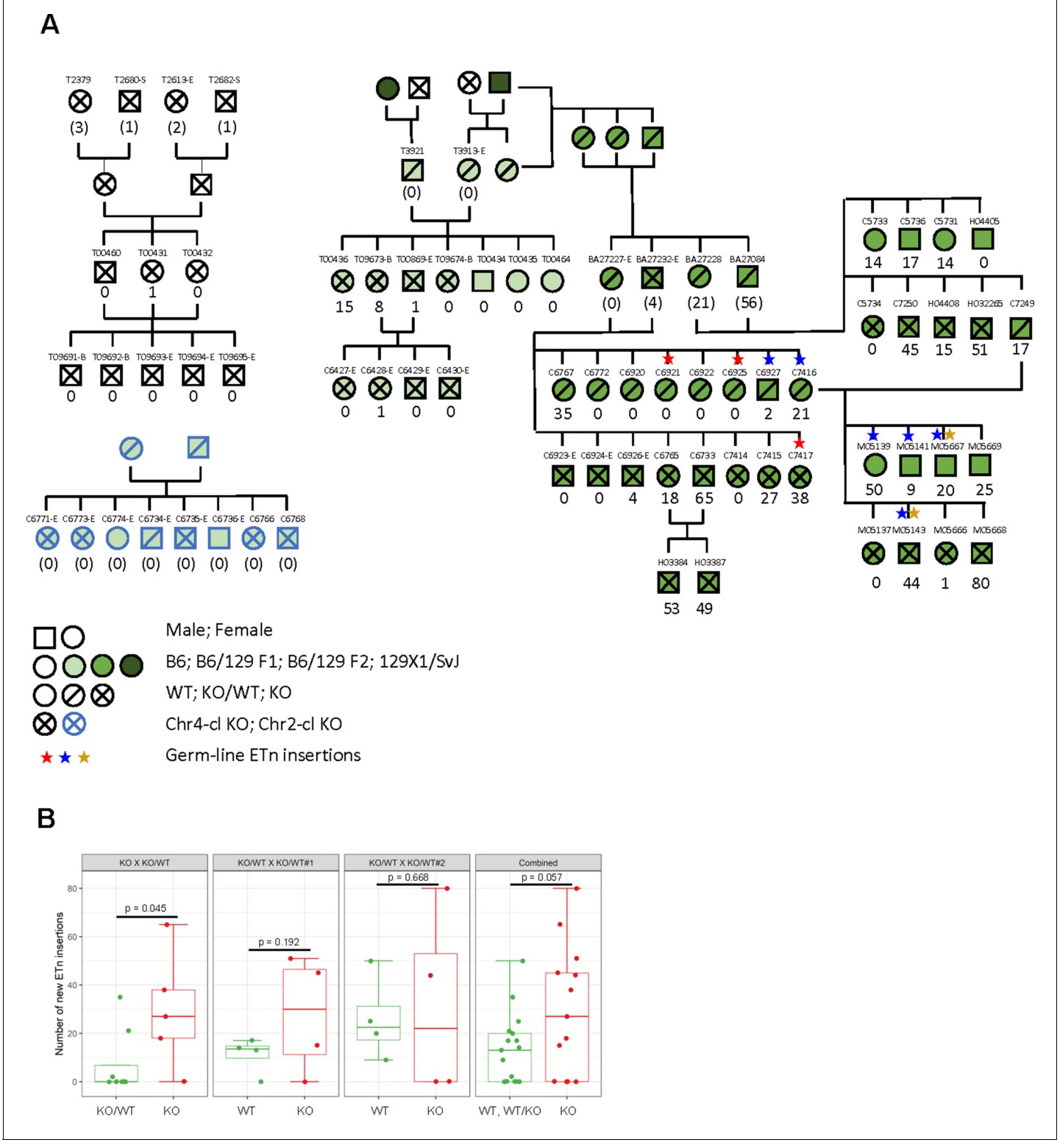

**Figure 4.** ETn retrotransposition in Chr4-cl KO mice. (**A**) Pedigree of mice used for transposon insertion screening by capture-seq in mice of different strain backgrounds. The number of novel ETn insertions (only present in one animal) are indicated. For animals whose direct ancestors have not been screened, the ETn insertions are shown in parentheses since parental inheritance cannot be excluded in that case. Germ line insertions are indicated by asterisks. All DNA samples were prepared from tail tissues unless noted (-S: spleen, -E: ear, -B:Blood) (**B**) Statistical analysis of ETn insertion frequency in tail tissue from 30 Chr4-cl KO, KO/WT and WT mice that were derived from one Chr4-c KO x KO/WT and two Chr4-cl KO/WT x KO/WT matings.

*Figure 4 continued on next page*

*Figure 4 continued*

Only DNA samples that were collected from juvenile tails were considered for this analysis. P-values were calculated using one-sided Wilcoxon Rank Sum Test. In the last panel, KO, WT and KO/WT mice derived from all matings were combined for the statistical analysis.

The online version of this article includes the following source data and figure supplement(s) for figure 4:

**Source data 1.** Coordinates of identified novel ETn insertions and supporting capture-seq read counts.
**Source data 2.** Sequences of capture-seq probes used to enrich genomic DNA for ETn and MuLV (RLTR4) insertions.
**Figure supplement 1.** Birth statistics of KRAB-ZFP cluster KO mice and TE reactivation in adult tissues.
**Figure supplement 2.** Identification of polymorphic ETn and MuLV retrotransposon insertions in Chr4-cl KO and WT mice.
**Figure supplement 3.** Confirmation of novel ETn insertions identified by capture-seq.

---

*figure supplement 3B*). In contrast, virtually all novel insertions that were only found in one animal were supported by significantly fewer reads (*Figure 4—figure supplement 3B*). This indicates that these elements resulted from retrotransposition events in the developing embryo and not in the zygote or parental germ cells. Indeed, we detected different sets of insertions in various tissues from the same animal (*Figure 4—figure supplement 3C*). Even between tail samples that were collected from the same animal at different ages, only a fraction of the new insertions were present in both samples, while technical replicates from the same genomic DNA samples showed a nearly complete overlap in insertions (*Figure 4—figure supplement 3D*).

Finally, we asked whether there were more novel ETn insertions in mice lacking the Chr4-cl relative to their wild type and heterozygous littermates in our pedigree. Interestingly, only one out of the eight Chr4-cl KO mice in a pure C57BL/6 strain background and none of the eight offspring from a Chr2-cl mating carried a single novel ETn insertion (*Figure 4A*). When crossing into a 129Sv background for a single generation before intercrossing heterozygous mice (F1), we observed 4 out of 8 Chr4-cl KO mice that contained at least one new ETn insertion, whereas none of 3 heterozygous mice contained any insertions. After crossing to the 129Sv background for a second generation (F2), we determined the number of novel ETn insertions in the offspring of one KO/WT x KO and two KO/WT x KO/WT matings, excluding all samples that were not derived from juvenile tail tissue. Only in the offspring of the KO/WT x KO mating, we observed a statistically significant higher average number of ETn insertions in KO vs. KO/WT animals (7.3 vs. 29.6, p=0.045, *Figure 4B*). Other than that, only a non-significant trend towards greater average numbers of ETn insertions in KO (11 vs. 27.8, p=0.192, *Figure 4B*) was apparent in one of the WT/KO x KO/WT matings whereas no difference in ETn insertion numbers between WT and KO mice could be observed in the second mating WT/KO x KO/WT (26 vs. 31, p=0.668, *Figure 4B*). When comparing all KO with all WT and WT/KO mice from these three matings, a trend towards more ETn insertions in KO remained but was not supported by strong significance (26 vs. 13, p=0.057, *Figure 4B*). Altogether, we observed a high variability in the number of new ETn insertions in both KO and WT but our data suggest that the Chr4-cl KRAB-ZFPs may have a modest effect on ETn retrotransposition rates in some mouse strains but other genetic and epigenetic effects clearly also play an important role.

## Discussion

C2H2 zinc finger proteins, about half of which contain a KRAB repressor domain, represent the largest DNA-binding protein family in mammals. Nevertheless, most of these factors have not been investigated using loss-of-function studies. The most comprehensive characterization of human KRAB-ZFPs revealed a strong preference to bind TEs (*Imbeault et al., 2017*; *Najafabadi et al., 2015*) yet their function remains unknown. In humans, very few TEs are capable of retrotransposition yet many of them, often tens of million years old, are bound by KRAB-ZFPs. While this suggests that human KRAB-ZFPs mainly serve to control TE-borne enhancers and may have potentially transcription-independent functions, we were interested in the biological significance of KRAB-ZFPs in restricting potentially active TEs. The mouse is an ideal model for such studies since the mouse genome contains several active TE families, including IAP, ETn and L1 elements. We found that many of the young KRAB-ZFPs present in the

genomic clusters of KRAB-ZFPs on chromosomes 2 and 4, which are highly expressed in a restricted pattern in ES cells, bound redundantly to these three active TE families. In several cases, KRAB-ZFPs bound to functionally constrained sequence elements we and others have demonstrated to be necessary for retrotransposition, including PBS and viral packaging signals. Targeting such sequences may help the host defense system keep pace with rapidly evolving mouse transposons. This provides strong evidence that many young KRAB-ZFPs are indeed expanding in response to TE activity. But do these young KRAB-ZFP genes limit the mobilization of TEs? Despite the large number of polymorphic ETn elements in mouse strains (*Nellåker et al., 2012*) and several reports of phenotype-causing novel ETn germ line insertions, no new ETn insertions were reported in recent screens of C57BL/6 mouse genomes (*Richardson et al., 2017*; *Gagnier et al., 2019*), indicating that the overall rate of ETn germ line mobilization in inbred mice is rather low. We have demonstrated that Chr4-cl KRAB-ZFPs control ETn/ETnERV expression in ES cells, but this does not lead to widespread ETn mobility in viable C57BL/6 mice. In contrast, we found numerous novel, including several germ line, ETn insertions in both WT and Chr4-cl KO mice in a C57BL/6 129Sv mixed genetic background, with generally more insertions in KO mice and in mice with more 129Sv DNA. This is consistent with a report detecting ETn insertions in FVB.129 mice (*Schauer et al., 2018*). Notably, there was a large variation in the number of new insertions in these mice, possibly caused by hyperactive polymorphic ETn insertions that varied from individual to individual, epigenetic variation at ETn insertions between individuals and/or the general stochastic nature of ETn mobilization. Furthermore, recent reports have suggested that KRAB-ZFP gene content is distinct in different strains of laboratory mice (*Lilue et al., 2018*; *Treger et al., 2019*), and reduced KRAB-ZFP gene content could contribute to increased activity in individual mice. Although we have yet to find obvious phenotypes in the mice carrying new insertions, novel ETn germ line insertions have been shown to cause phenotypes from short tails (*Lugani et al., 2013*; *Semba et al., 2013*; *Vlangos et al., 2013*) to limb malformation (*Kano et al., 2007*) and severe morphogenetic defects including polypodia (*Lehoczky et al., 2013*) depending upon their insertion site.

Despite a lack of widespread ETn activation in Chr4-cl KO mice, it still remains to be determined whether other TEs, like L1, IAP or other LTR retrotransposons are activated in any of the KRAB-ZFP cluster KO mice, which will require the development of additional capture-seq based assays. Notably, two of the heterozygous matings from Chr2-cl KO mice failed to produce viable knockout offspring, which could indicate a TE-reactivation phenotype. It may also be necessary to generate compound homozygous mutants of distinct KRAB-ZFP clusters to eliminate redundancy before TEs become unleashed. The KRAB-ZFP cluster knockouts produced here will be useful reagents to test such hypotheses. In sum, our data supports that a major driver of KRAB-ZFP gene expansion in mice is recent retrotransposon insertions, and that redundancy within the KRAB-ZFP gene family and with other TE restriction pathways provides protection against widespread TE mobility, explaining the non-essential function of the majority of KRAB-ZFP genes.

## Materials and methods

**Key resources table**

| Reagent type (species) or resource | Designation | Source or reference | Identifiers | Additional information |
|---|---|---|---|---|
| Strain, strain background (*Mus musculus*) | 129 × 1/SvJ | The Jackson Laboratory | 000691 | Mice used to generate mixed strain Chr4-cl KO mice |
| Cell line (*Homo-sapiens*) | HeLa | ATCC | ATCC CCL-2 | |
| Cell line (*Mus musculus*) | JM8A3.N1 C57BL/6N-Atm1Brd | KOMP Repository | PL236745 | B6 ES cells used to generate KO cell lines and mice |

*Continued on next page*

*Continued*

| Reagent type (species) or resource | Designation | Source or reference | Identifiers | Additional information |
|---|---|---|---|---|
| Cell line (*Mus musculus*) | B6;129- Gt(ROSA) 26Sortm1 (cre/ERT)Nat/J | The Jackson Laboratory | 004847 | ES cells used to generate KO cell lines and mice |
| Cell line (*Mus musculus*) | R1 ES cells | Andras Nagy lab | R1 | 129 ES cells used to generate KO cell lines and mice |
| Cell line (*Mus musculus*) | F9 Embryonic carcinoma cells | ATCC | ATCC CRL-1720 | |
| Antibody | Mouse monoclonal ANTI-FLAG M2 antibody | Sigma-Aldrich | Cat# F1804, RRID:AB_262044 | ChIP (1 µg/$10^7$ cells) |
| Antibody | Rabbit polyclonal anti-HA | Abcam | Cat# ab9110, RRID:AB_307019 | ChIP (1 µg/$10^7$ cells) |
| Antibody | Mouse monoclonal anti-HA | Covance | Cat# MMS-101P-200, RRID:AB_10064068 | |
| Antibody | Rabbit polyclonal anti-H3K9me3 | Active Motif | Cat# 39161, RRID:AB_2532132 | ChIP (3 µl/$10^7$ cells) |
| Antibody | Rabbit polyclonal anti-GFP | Thermo Fisher Scientific | Cat# A-11122, RRID:AB_221569 | ChIP (1 µg/$10^7$ cells) |
| Antibody | Rabbit polyclonal anti- H3K4me3 | Abcam | Cat# ab8580, RRID:AB_306649 | ChIP (1 µg/$10^7$ cells) |
| Antibody | Rabbit polyclonal anti- H3K4me1 | Abcam | Cat# ab8895, RRID:AB_306847 | ChIP (1 µg/$10^7$ cells) |
| Antibody | Rabbit polyclonal anti- H3K27ac | Abcam | Cat# ab4729, RRID:AB_2118291 | ChIP (1 µg/$10^7$ cells) |
| Recombinant DNA reagent | pCW57.1 | Addgene | RRID:Addgene_41393 | Inducible lentiviral expression vector |
| Recombinant DNA reagent | pX330-U6-Chimeric_BB-CBh-hSpCas9 | Addgene | RRID:Addgene_42230 | CRISPR/Cas9 expression construct |
| Sequence-based reagent | Chr2-cl KO gRNA.1 | This paper | Cas9 gRNA | GCCGTTGCTCAGTCCAAATG |
| Sequenced-based reagent | Chr2-cl KO gRNA.2 | This paper | Cas9 gRNA | GATACCAGAGGTGGCCGCAAG |
| Sequenced-based reagent | Chr4-cl KO gRNA.1 | This paper | Cas9 gRNA | GCAAAGGGGCTCCTCGATGGA |
| Sequence-based reagent | Chr4-cl KO gRNA.2 | This paper | Cas9 gRNA | GTTTATGGCCGTGCTAAGGTC |
| Sequenced-based reagent | Chr10-cl KO gRNA.1 | This paper | Cas9 gRNA | GTTGCCTTCATCCCACCGTG |
| Sequenced-based reagent | Chr10-cl KO gRNA.2 | This paper | Cas9 gRNA | GAAGTTCGACTTGGACGGGCT |
| Sequenced-based reagent | Chr13.1-cl KO gRNA.1 | This paper | Cas9 gRNA | GTAACCCATCATGGGCCCTAC |
| Sequenced-based reagent | Chr13.1-cl KO gRNA.2 | This paper | Cas9 gRNA | GGACAGGTTATAGGTTTGAT |
| Sequenced-based reagent | Chr13.2-cl KO gRNA.1 | This paper | Cas9 gRNA | GGGTTTCTGAGAAACGTGTA |

*Continued on next page*

*Continued*

| Reagent type (species) or resource | Designation | Source or reference | Identifiers | Additional information |
|---|---|---|---|---|
| Sequenced-based reagent | Chr13.2-cl KO gRNA.2 | This paper | Cas9 gRNA | GTGTAATGAGT TCTTATATC |
| Commercial assay or kit | SureSelectQXT Target Enrichment kit | Agilent | G9681-90000 | |
| Software, algorithm | Bowtie | http://bowtie-bio.sourceforge.net | RRID:SCR_005476 | |
| Software, algorithm | MACS14 | https://bio.tools/macs | RRID:SCR_013291 | |
| Software, algorithm | Tophat | https://ccb.jhu.edu | RRID:SCR_013035 | |

## Cell lines and transgenic mice

Mouse ES cells and F9 EC cells were cultivated as described previously (*Wolf et al., 2015b*) unless stated otherwise. Chr4-cl KO ES cells originate from B6;129- Gt(ROSA)26Sortm1(cre/ERT)Nat/J mice (Jackson lab), all other KRAB-ZFP cluster KO ES cell lines originate from JM8A3.N1 C57BL/6N-A^tm1Brd ES cells (KOMP Repository). Chr2-cl KO and WT ES cells were initially grown in serum-containing media (*Wolf et al., 2015b*) but changed to 2i media (*De Iaco et al., 2017*) for several weeks before analysis. To generate Chr4-cl and Chr2-cl KO mice, the cluster deletions were repeated in B6 ES (KOMP repository) or R1 (Nagy lab) ES cells, respectively, and heterozygous clones were injected into B6 albino blastocysts. Chr2-cl KO mice were therefore kept on a mixed B6/Svx129/Sv-CP strain background while Chr4-cl KO mice were initially derived on a pure C57BL/6 background. For capture-seq screens, Chr4-cl KO mice were crossed with 129 × 1/SvJ mice (Jackson lab) to produce the founder mice for Chr4-cl KO and WT (B6/129 F1) offspring. Chr4-cl KO/WT (B6/129 F1) were also crossed with 129 × 1/SvJ mice to get Chr4-cl KO/WT (B6/129 F1) mice, which were intercrossed to give rise to the parents of Chr4-cl KO/KO and KO/WT (B6/129 F2) offspring.

## Generation of KRAB-ZFP expressing cell lines

KRAB-ZFP ORFs were PCR-amplified from cDNA or synthesized with codon-optimization (*Supplementary file 1*), and stably expressed with 3XFLAG or 3XHA tags in F9 EC or ES cells using *Sleeping beauty* transposon-based (*Wolf et al., 2015b*) or lentiviral expression vectors (*Imbeault et al., 2017*; *Supplementary file 1*). Cells were selected with puromycin (1 µg/ml) and resistant clones were pooled and further expanded for ChIP-seq.

## CRISPR/Cas9 mediated deletion of KRAB-ZFP clusters and an MMETn insertion

All gRNAs were expressed from the pX330-U6-Chimeric_BB-CBh-hSpCas9 vector (RRID:Addgene_42230) and nucleofected into $10^6$ ES cells using Amaxa nucleofection in the following amounts: 5 µg of each pX330-gRNA plasmid, 1 µg pPGK-puro and 500 pmoles single-stranded repair oligos (*Supplementary file 3*). One day after nucleofection, cells were kept under puromycin selection (1 µg/ml) for 24 hr. Individual KO and WT clones were picked 7–8 days after nucleofection and expanded for PCR genotyping (*Supplementary file 3*).

## ChIP-seq analysis

For ChIP-seq analysis of KRAB-ZFP expressing cells, 5–10 × $10^7$ cells were crosslinked and immuno-precipitated with anti-FLAG (Sigma-Aldrich Cat# F1804, RRID:AB_262044) or anti-HA (Abcam Cat# ab9110, RRID:AB_307019 or Covance Cat# MMS-101P-200, RRID:AB_10064068) antibody using one of two previously described protocols (*O'Geen et al., 2010*; *Imbeault et al., 2017*) as indicated in *Supplementary file 1*. H3K9me3 distribution in Chr4-cl, Chr10-cl, Chr13.1-cl and Chr13.2-cl KO ES cells was determined by native ChIP-seq with anti-H3K9me3 serum (Active Motif Cat# 39161, RRID:AB_2532132) as described previously (*Karimi et al., 2011*). In Chr2-cl KO ES cells, H3K9me3 and

KAP1 ChIP-seq was performed as previously described (*Ecco et al., 2016*). In Chr4-cl KO and WT ES cells KAP1 binding was determined by endogenous tagging of KAP1 with C-terminal GFP (*Supplementary file 3*), followed by FACS to enrich for GFP-positive cells and ChIP with anti-GFP (Thermo Fisher Scientific Cat# A-11122, RRID:AB_221569) using a previously described protocol (*O'Geen et al., 2010*). For ChIP-seq analysis of active histone marks, cross-linked chromatin from ES cells or testis (from two-week old mice) was immunoprecipitated with antibodies against H3K4me3 (Abcam Cat# ab8580, RRID:AB_306649), H3K4me1 (Abcam Cat# ab8895, RRID:AB_306847) and H3K27ac (Abcam Cat# ab4729, RRID:AB_2118291) following the protocol developed by *O'Geen et al., 2010* or *Khil et al., 2012* respectively.

ChIP-seq libraries were constructed and sequenced as indicated in *Supplementary file 4*. Reads were mapped to the mm9 genome using Bowtie (RRID:SCR_005476; settings: –best) or Bowtie2 (*Langmead and Salzberg, 2012*) as indicated in *Supplementary file 4*. Under these settings, reads that map to multiple genomic regions are assigned to the top-scored match and, if a set of equally good choices is encountered, a pseudo-random number is used to choose one location. Peaks were called using MACS14 (RRID:SCR_013291) under high stringency settings (p<1e-10, peak enrichment >20) (*Zhang et al., 2008*). Peaks were called both over the Input control and a FLAG or HA control ChIP (unless otherwise stated in *Supplementary file 4*) and only peaks that were called in both settings were kept for further analysis. In cases when the stringency settings did not result in at least 50 peaks, the settings were changed to medium (p<1e-10, peak enrichment >10) or low (p<1e-5, peak enrichment >10) stringency (*Supplementary file 4*). For further analysis, all peaks were scaled to 200 bp regions centered around the peak summits. The overlap of the scaled peaks to each repeat element in UCSC Genome Browser (RRID:SCR_005780) were calculated by using the bedfisher function (settings: -f 0.25) from BEDTools (RRID:SCR_006646). The right-tailed p-values between pair-wise comparison of each ChIP-seq peak and repeat element were extracted, and then adjusted using the Benjamini-Hochberg approach implemented in the R function p.adjust(). Binding motifs were determined using only nonrepetitive (<10% repeat content) peaks with MEME (*Bailey et al., 2009*). MEME motifs were compared with in silico predicted motifs (*Najafabadi et al., 2015*) using Tomtom (*Bailey et al., 2009*) and considered as significantly overlapping with a False Discovery Rate (FDR) below 0.1. To find MEME and predicted motifs in repetitive peaks, we used FIMO (*Bailey et al., 2009*). Differential H3K9me3 and KAP1 distribution in WT and Chr2-cl or Chr4-cl KO ES cells at TEs was determined by counting ChIP-seq reads overlapping annotated insertions of each TE group using BEDTools (MultiCovBed). Additionally, ChIP-seq reads were counted at the TE fraction that was bound by Chr2-cl or Chr4-cl KRAB-ZFPs (overlapping with 200 bp peaks). Count tables were concatenated and analyzed using DESeq2 (*Love et al., 2014*). The previously published ChIP-seq datasets for KAP1 (*Castro-Diaz et al., 2014*) and H3K9me3 (*Dan et al., 2014*) were re-mapped using Bowtie (–best).

## Luciferase reporter assays

For KRAB-ZFP repression assays, double-stranded DNA oligos containing KRAB-ZFP target sequences (*Supplementary file 3*) were cloned upstream of the SV40 promoter of the pGL3-Promoter vector (Promega) between the restriction sites for NheI and XhoI. 33 ng of reporter vectors were co-transfected (Lipofectamine 2000, Thermofisher) with 33 ng pRL-SV40 (Promega) for normalization and 33 ng of transient KRAB-ZFP expression vectors (in pcDNA3.1) or empty pcDNA3.1 into 293 T cells seeded one day earlier in 96-well plates. Cells were lysed 48 hr after transfection and luciferase/Renilla luciferase activity was measured using the Dual-Luciferase Reporter Assay System (Promega). To measure the transcriptional activity of the MMETn element upstream of the *Cd59a* gene, fragments of varying sizes (*Supplementary file 3*) were cloned into the promoter-less pGL3-basic vector (Promega) using NheI and NcoI sites. 70 ng of reporter vectors were cotransfected with 30 ng pRL-SV40 into feeder-depleted Chr4-cl WT and KO ES cells, seeded into a gelatinized 96-well plate 2 hr before transfection. Luciferase activity was measured 48 hr after transfection as described above.

## RNA-seq analysis

Whole RNA was purified using RNeasy columns (Qiagen) with on column DNase treatment or the High Pure RNA Isolation Kit (Roche) (*Supplementary file 4*). Tissues were first lysed in TRIzol reagent

(ThermoFisher) and RNA was purified after the isopropanol precipitation step using RNeasy columns (Qiagen) with on column DNase treatment. Libraries were generated using the SureSelect Strand-Specific RNA Library Prep kit (Agilent) or Illumina's TruSeq RNA Library Prep Kit (with polyA selection) and sequenced as 50 or 100 bp paired-end reads on an Illumina HiSeq2500 (RRID:SCR_016383) or HiSeq3000 (RRID:SCR_016386) machine (*Supplementary file 4*). RNA-seq reads were mapped to the mouse genome (mm9) using Tophat (RRID:SCR_013035; settings: –I 200000 g 1) unless otherwise stated. These settings allow each mappable read to be reported once, in case the read maps to multiple locations equally well, one match is randomly chosen. For differential transposon expression, mapped reads that overlap with TEs annotated in Repeatmasker (RRID:SCR_012954) were counted using BEDTools MultiCovBed (setting: -split). Reads mapping to multiple fragments that belong to the same TE insertion (as indicated by the repeat ID) were summed up. Only transposons with a total of at least 20 (for two biological replicates) or 30 (for three biological replicates) mapped reads across WT and KO samples were considered for differential expression analysis. Transposons within the deleted KRAB-ZFP cluster were excluded from the analysis. Read count tables were used for differential expression analysis with DESeq2 (RRID:SCR_015687). For differential gene expression analysis, reads overlapping with gene exons were counted using HTSeq-count and analyzed using DESeq2. To test if KRAB-ZFP peaks are significantly enriched near up- or down-regulated genes, a binomial test was performed. Briefly, the proportion of the peaks that are located within a certain distance up- or downstream to the TSS of genes was determined using the windowBed function of BED tools. The probability $p$ in the binomial distribution was estimated as the fraction of all genes overlapped with KRAB-ZFP peaks. Then, given $n$ which is the number of specific groups of genes, and $x$ which is the number of this group of genes overlapped with peaks, the R function binom.test() was used to estimate the p-value based on right-tailed Binomial test. Finally, the adjusted p-values were determined separately for LTR and LINE retrotransposon groups using the Benjamini-Hochberg approach implemented in the R function p.adjust().

## Reduced representation bisulfite sequencing (RRBS-seq)

For RRBS-seq analysis, Chr4-cl WT and KO ES cells were grown in either standard ES cell media containing FCS or for one week in 2i media containing vitamin C as described previously (*Blaschke et al., 2013*). Genomic DNA was purified from WT and Chr4-cl KO ES cells using the Quick-gDNA purification kit (Zymo Research) and bisulfite-converted with the NEXTflex Bisulfite-Seq Kit (Bio Scientific) using Msp1 digestion to fragment DNA. Libraries were sequenced as 50 bp paired-end reads on an Illumina HiSeq. The reads were processed using Trim Galore (–illumina – paired –rrbs) to trim poor quality bases and adaptors. Additionally, the first 5 nt of R2 and the last 3 nt of R1 and R2 were trimmed. Reads were then mapped to the reference genome (mm9) using Bismark (*Krueger and Andrews, 2011*) to extract methylation calling results. The CpG methylation pattern for each covered CpG dyads (two complementary CG dinucleotides) was calculated using a custom script (*Source code 1*: get_CpG_ML.pl). For comparison of CpG methylation between WT and Chr4-cl KO ES cells (in serum or 2i + Vitamin C conditions) only CpG sites with at least 10-fold coverage in each sample were considered for analysis.

## Retrotransposition assay

The retrotransposition vectors pCMV-MusD2, pCMV-MusD2-neoTNF and pCMV-ETnI1-neoTNF (*Ribet et al., 2004*) were a kind gift from Dixie Mager. To partially delete the Gm13051 binding site within pCMV-MusD2-neoTNF, the vector was cut with KpnI and re-ligated using a repair oligo, leaving a 24 bp deletion within the Gm13051 binding site. The Rex2 binding site in pCMV-ETnI1-neoTNF was deleted by cutting the vector with EcoRI and XbaI followed by re-ligation using two overlapping PCR products, leaving a 45 bp deletion while maintaining the rest of the vector unchanged (see *Supplementary file 3* for primer sequences). For MusD retrotransposition assays, $5 \times 10^4$ HeLa cells (ATCC CCL-2) were transfected in a 24-well dish with 100 ng pCMV-MusD2-neoTNF or pCMV-MusD2-neoTNF (ΔGm13051-m) using Lipofectamine 2000. For ETn retrotransposition assays, 50 ng of pCMV-ETnI1-neoTNF or pCMV-ETnI1-neoTNF (ΔRex2) vectors were cotransfected with 50 ng pCMV-MusD2 to provide gag and pol proteins in trans. G418 (0.6 mg/ml) was added five days after transfection and cells were grown under selection until colonies were readily visible by eye. G418-resistant colonies were stained with Amido Black (Sigma).

## Capture-seq screen

To identify novel retrotransposon insertions, genomic DNA from various tissues (*Supplementary file 4*) was purified and used for library construction with target enrichment using the SureSelect<sup>QXT</sup> Target Enrichment kit (Agilent). Custom RNA capture probes were designed to hybridize with the 120 bp 5' ends of the 5' LTRs and the 120 bp 3' ends of the 3' LTR of about 600 intact (internal region flanked by two LTRs) MMETn/RLTRETN retrotransposons or of 140 RLTR4_MM/RLTR4 retrotransposons that were upregulated in Chr4-cl KO ES cells (*Figure 4—source data 2*). Enriched libraries were sequenced on an Illumina HiSeq as paired-end 50 bp reads. R1 and R2 reads were mapped to the mm9 genome separately, using settings that only allow non-duplicated, uniquely mappable reads (Bowtie -m 1 –`best` –`strata`; samtools rmdup -s) and under settings that allow multimapping and duplicated reads (Bowtie –`best`). Of the latter, only reads that overlap (min. 50% of read) with RLTRETN, MMETn-int, ETnERV-int, ETnERV2-int or ETnERV3-int repeats (ETn) or RLTR4, RLTR4_MM-int or MuLV-int repeats (RLTR4) were kept. Only uniquely mappable reads whose paired reads were overlapping with the repeats mentioned above were used for further analysis. All ETn- and RLTR4-paired reads were then clustered (as bed files) using BEDTools (bedtools merge -i -n -d 1000) to receive a list of all potential annotated and non-annotated new ETn or RLTR4 insertion sites and all overlapping ETn- or RLTR4-paired reads were counted for each sample at each locus. Finally, all regions that were located within 1 kb of an annotated RLTRETN, MMETn-int, ETnERV-int, ETnERV2-int or ETnERV3-int repeat as well as regions overlapping with previously identified polymorphic ETn elements (*Nellåker et al., 2012*) were removed. Genomic loci with at least 10 reads per million unique ETn- or RLTR4-paired reads were considered as insertion sites. To qualify for a de-novo insertion, we allowed no called insertions in any of the other screened mice at the locus and not a single read at the locus in the ancestors of the mouse. Insertions at the same locus in at least two siblings from the same offspring were considered as germ line insertions, if the insertion was absent in the parents and mice who were not direct descendants from these siblings. Full-length sequencing of new ETn insertions was done by Sanger sequencing of short PCR products in combination with Illumina sequencing of a large PCR product (*Supplementary file 3*), followed by de-novo assembly using the Unicycler software.

## Acknowledgements

We thank Alex Grinberg, Jeanne Yimdjo and Victoria Carter for generating and maintaining transgenic mice. We also thank members of the Macfarlan and Trono labs for useful discussion, Steven Coon, James Iben, Tianwei Li and Anna Malawska for NGS and computational support. This work was supported by NIH grant 1ZIAHD008933 and the NIH DDIR Innovation Award program (TSM), and by subsidies from the Swiss National Science Foundation (310030_152879 and 310030B_173337) and the European Research Council (KRABnKAP, No. 268721; Transpos-X, No. 694658) (DT).

## Additional information

### Funding

| Funder | Grant reference number | Author |
| --- | --- | --- |
| Eunice Kennedy Shriver National Institute of Child Health and Human Development | 1ZIAHD008933 | Todd S Macfarlan |
| Swiss National Science Foundation | 310030_152879 | Didier Trono |
| Swiss National Science Foundation | 310030B_173337 | Didier Trono |
| European Research Council | No. 268721 | Didier Trono |
| European Research Council | No 694658 | Didier Trono |

The funders had no role in study design, data collection and interpretation, or the decision to submit the work for publication.

## Author contributions

Gernot Wolf, Conceptualization, Data curation, Formal analysis, Investigation, Methodology, Writing - original draft; Alberto de Iaco, Conceptualization, Data curation, Formal analysis, Investigation, Methodology, Writing - original draft, Writing - review and editing; Ming-An Sun, Conceptualization, Data curation, Software, Formal analysis, Investigation, Visualization, Methodology, Writing - review and editing; Melania Bruno, Conceptualization, Formal analysis, Investigation, Writing - review and editing; Matthew Tinkham, Don Hoang, Sherry Ralls, Investigation; Apratim Mitra, Data curation, Software, Formal analysis, Visualization; Didier Trono, Conceptualization, Resources, Supervision, Funding acquisition, Investigation, Writing - review and editing; Todd S Macfarlan, Conceptualization, Resources, Supervision, Funding acquisition, Investigation, Methodology, Writing - original draft, Project administration, Writing - review and editing

## Author ORCIDs

Gernot Wolf (iD) https://orcid.org/0000-0002-3943-8662
Melania Bruno (iD) http://orcid.org/0000-0002-8401-7744
Didier Trono (iD) http://orcid.org/0000-0002-3383-0401
Todd S Macfarlan (iD) https://orcid.org/0000-0003-2495-9809

## Ethics

Animal experimentation: All studies using mice were performed in accordance to the Guide for the Care and Use of Laboratory Animals of the NIH, under IACUC animal protocol (ASP )18-026.

## Decision letter and Author response

Decision letter https://doi.org/10.7554/eLife.56337.sa1
Author response https://doi.org/10.7554/eLife.56337.sa2

# Additional files

## Supplementary files

• Source code 1. Custom Perl script used to get methylation pattern for each CpG dyads from Bismark methylation calling results.

• Supplementary file 1. Experimental parameters, gene-centered informa347tion and summary of KRAB-ZFP ChIP-seq analysis.

• Supplementary file 2. Differential CpG methylation status of TEs in WT and Chr4-cl KO ES cells in serum and 2i culture conditions.

• Supplementary file 3. Sequence information of used PCR primers, gRNAs and cloned oligos for luciferase repression assays.

• Supplementary file 4. Overview of generated NGS data.

• Transparent reporting form

## Data availability

All NGS data has been deposited in GEO (GSE115291). Sequences of full-length de novo ETn insertions have been deposited in the GenBank database (MH449667- MH449669).

The following datasets were generated:

| Author(s) | Year | Dataset title | Dataset URL | Database and Identifier |
|---|---|---|---|---|
| Wolf G | 2019 | Retrotransposon reactivation and mobilization upon deletions of megabase scale KRAB zinc finger | https://www.ncbi.nlm.nih.gov/geo/query/acc.cgi?acc=GSE115291 | NCBI Gene Expression Omnibus, GSE115291 |

| Author(s) | Year | Dataset title | Dataset URL | Database and Identifier |
|---|---|---|---|---|
| | | gene clusters in mice | | |
| Wolf G | 2019 | Mus musculus musculus strain C57BL/6x129X1/SvJ retrotransposon MMETn-int, complete sequence | https://www.ncbi.nlm.nih.gov/nuccore/MH449667 | NCBI GenBank, MH449667 |
| Wolf G | 2019 | Mus musculus musculus strain C57BL/6x129X1/SvJ retrotransposon MMETn-int, complete sequence | https://www.ncbi.nlm.nih.gov/nuccore/MH449668 | NCBI GenBank, MH449668 |
| Wolf G | 2019 | Mus musculus musculus strain C57BL/6x129X1/SvJ retrotransposon MMETn-int, complete sequence | https://www.ncbi.nlm.nih.gov/nuccore/MH449669 | NCBI GenBank, MH449669 |

The following previously published datasets were used:

| Author(s) | Year | Dataset title | Dataset URL | Database and Identifier |
|---|---|---|---|---|
| Castro-Diaz N, Ecco G, Coluccio A, Kapopoulou A, Duc J, Trono D | 2014 | Evollutionally dynamic L1 regulation in embryonic stem cells | https://www.ncbi.nlm.nih.gov/geo/query/acc.cgi?acc=GSE58323 | NCBI Gene Expression Omnibus, GSM1406445 |
| Andrew ZX | 2014 | H3K9me3_ChIPSeq (Ctrl) | https://www.ncbi.nlm.nih.gov/geo/query/acc.cgi?acc=GSM1327148 | NCBI Gene Expression Omnibus, GSM1327148 |

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
