## [Decision Letter]

**Acceptance summary:**

This is the first study that addresses in vivo the targets and function of genomic clusters of KRAB-ZFP-encoding genes. This family of proteins reveals the complexity, redundancy and specificity (of targets and of cellular type of expression) of the evolutionary processes that control ever adapting retrotransposons.

**Decision letter after peer review:**

Thank you for submitting your article "KRAB-zinc finger protein gene expansion in response to active retrotransposons in the murine lineage" for consideration by *eLife*. Your article has been reviewed by three peer reviewers, including Deborah Bourc'his as the Reviewing Editor and Reviewer #1, and the evaluation has been overseen by George Perry as the Senior Editor. The following individual involved in review of your submission has agreed to reveal their identity: Joan Barau (Reviewer #3).

The reviewers have discussed the reviews with one another and the Reviewing Editor has drafted this decision to help you prepare a revised submission.

Summary:

Restriction of transposable elements relies on multiple pathways, among which KRAB-ZFP proteins play a key role through their diversity and fast evolvability. The repertoire of KRAB-ZFPs is highly species-specific, likely driven by the pressure to adapt to host retrotransposon load, although they can also be repurposed for other regulatory functions over evolutionary time. Their functional study is challenging, because of their redundant and complementary functions, and also because of their sequence similarity and common organization in large genomic clusters.

In this study, the authors probed the genomic targets and roles of a large set of KRAB-ZFPs that are specifically expressed in pluripotent cells and that have recently arisen in the mouse genome, which contains a large fraction of retrotransposons of LTR and non-LTR classes that are still potentially mobile. Using systematic expression of epitope-tagged KRAB-ZFP transgenes and deletions of large clusters in both embryonic stem cells and in mice, the authors demonstrate that: 1) mouse KRAB-ZFPs primarily bind to retrotransposons, 2) large KRAB-ZFP gene clusters are required for transcriptional repression of evolutionarily young retrotransposons (LINE, IAP, and ETn) in pluripotent cells, 3) large pluripotency-associated KRAB-ZFP gene clusters are mostly dispensable for viability and development, and 4) these same KRAB-ZFP clusters have very low effect on the mobility of some young LTR retrotransposons of the ETn/MusD class, although genetic background may play a role. Interbreeding between KO/KO animals over several generations may enhance mobilization and phenotypic outcomes, but this was not addressed here.

This is the first study of this scale to investigate KRAB-ZFP targets and function in mice. The approaches are state-of-the art, involving large, megabase-scale deletions, development of capture-sequencing for mapping de novo insertions of ETn/MusD copies, and analysis of the effects of KRAB-ZFP on retrotransposon mobilization at the organism level. The manuscript is clear and conclusions are fair. However, the search of new insertions in mutant mouse pedigrees is not fully conclusive, since the contribution of the genetic background on ETn/MusD mobilization seems to be higher than the effects of cluster deletions – if any.

Essential revisions:

1) Related to the identification of new insertions: the absence of basic reporting on the efficiency of the capture sequencing approach renders difficult to evaluate if the numbers provided are realistic or not. Recovery rate should be assessed using reference or parental retrotransposons. Also, a table with all insertions (genomic coordinates) and individuals could not be found and should be provided. PCR-validating a significant number of putative somatic insertions would be useful to estimate the rate of false positives, since these insertions have low-read counts (only germline insertions were tested in Figure 4—figure supplement 3A).

2) It is not very clear as to whether the retrotransposition rate is also low in ES cells. Could the authors be more clear about this? On this matter, the authors should test the rate of retrotransposition using the plasmid-borne ETn/MusD retrotransposition assay in the context of cl4 KO/KO ES cells (as used in Figure Sup3C). Similarly, they could use available LINE-1 and/or IAP retrotransposition cassettes in cl2 KO/KO ES cells, where these elements are the most reactivated (according to the RNA-seq data).

3) As most of the manuscript is heavily focused on ETn elements, the authors should provide more information on this family. Notably, the authors should make clear from the beginning that ETnERV = MusD, explain what MusD, ETnI and ETnII are -relative to each others- and give indications about the number of these different ETn-related elements in the mouse reference genome (C57bl6/J) genome and also in 129Sv.

4) Figure 2C: Was CpG methylation calculated over the entire sequence of full-length copies only? Or are the methylation values extracted from all ETn- or IAP-related reads ? This should be clarified. Also, the authors should repeat their analysis by specifically focusing on CpG methylation located in the LTR sequence of these elements. This is where DNA methylation has been shown to be targeted and to matter for retrotransposons repression. Changes may be greater.

5) The data graphed in Figure 4B do not seem to reflect the pedigree depicted in Figure 4A. Only 24 progenies could be counted for the KO/WT x KO/WT matings (12 WT/WT or WT/KO, and 12 KO/KO). In addition, some data points (such as the second from top, x=KO, y=~65) seem to originate from a different type of cross (heterozygote x homozygote). The overall conclusion (that KO produce on average more ETn insertions) could be quite different if a small number of individuals was misplaced in the graph, given the size of the samples. It is important to make sure which animals were accounted from the pedigree in 4A to produce the boxplot 4B.

---

## [Author Response]

Essential revisions:1) Related to the identification of new insertions: the absence of basic reporting on the efficiency of the capture sequencing approach renders difficult to evaluate if the numbers provided are realistic or not. Recovery rate should be assessed using reference or parental retrotransposons. Also, a table with all insertions (genomic coordinates) and individuals could not be found and should be provided. PCR-validating a significant number of putative somatic insertions would be useful to estimate the rate of false positives, since these insertions have low-read counts (only germline insertions were tested in Figure 4—figure supplement 3A).

We compiled a list of largely intact MMETN insertions (two RLTRETN_Mm LTRs flanking an internal MMETn-int region) in the mm9 reference genome and we identified >95% of these 309 insertions in the majority of samples. The average read number supporting these insertions was approximately two-fold higher than for the novel germ-line insertions that we identified. This difference is most likely a result of the heterozygosity of the novel insertions. We added this data to Figure 4—figure supplement 3B and refer to it in the main text.

As for the PCR validation, the five tested and confirmed insertions were actually putative somatic insertions (not germ-line insertions) with low read counts (Figure 4—figure supplement 3A). We only tested 5 somatic insertions for validation by PCR, and all 5 were confirmed. This suggests that our capture -seq pipeline has a low false positive rate.

We compiled a list of all novel insertions with supporting read numbers and added it as Figure 4—source data 1.

2) It is not very clear as to whether the retrotransposition rate is also low in ES cells. Could the authors be more clear about this? On this matter, the authors should test the rate of retrotransposition using the plasmid-borne ETn/MusD retrotransposition assay in the context of cl4 KO/KO ES cells (as used in Figure Sup3C). Similarly, they could use available LINE-1 and/or IAP retrotransposition cassettes in cl2 KO/KO ES cells, where these elements are the most reactivated (according to the RNA-seq data).

We screened the two KO/WT ES cell lines that had been used to derive the Chr4-cl KO mouse line by Capture-seq to exclude that newly identified insertions had been present in the founder ES cells already. We could not identify any new insertions in these cell lines either; however, we did not screen KO/KO ES cells so we cannot speculate about retrotransposition rate of endogenous ETn elements in WT and KO ES cells.

We attempted repeatedly to perform our retrotransposition assays in Chr4-cl WT and KO ES cells using ETn and MusD reporters, yet were not able to observe resistant colonies that would indicate retrotransposition events in any of the ES cell clones (WT or KO). Our explanation is that KRAB-ZFP independent mechanisms restrict retrotransposition in ES cells. The transfection efficiency is also lower in ES cells which could contribute to reduced rates relative to other cultured cell lines.

As for Chr2-cl KO ES cells, such experiments are, together with Capture-seq screens of Chr2-cl KO mice on the way but are planned to be included in a follow-up study that focuses on IAP and L1 retrotransposition.

3) As most of the manuscript is heavily focused on ETn elements, the authors should provide more information on this family. Notably, the authors should make clear from the beginning that ETnERV = MusD, explain what MusD, ETnI and ETnII are -relative to each others- and give indications about the number of these different ETn-related elements in the mouse reference genome (C57bl6/J) genome and also in 129Sv.

We have added the following to the manuscript:

“including the closely related MMETn (hereafter referred to as ETn) and ETnERV (also known as MusD) elements (Figure 1A and Figure 1—source data 2). ETn elements are non-autonomous LTR retrotransposons that require trans-complementation by the fully coding ETnERV elements that contain Gag Pro and Pol genes Ribet et al., 2004().These elements have accumulated to ~240 and ~100 copies in the reference C57BL/6 genome, respectively, with ~550 solitary LTRs Baust et al., 2003(). Both ETn and ETnERVs are still active, generating polymorphisms and mutations in several mouse strains Gagnier et al., 2019().”

4) Figure 2C: Was CpG methylation calculated over the entire sequence of full-length copies only? Or are the methylation values extracted from all ETn- or IAP-related reads ? This should be clarified. Also, the authors should repeat their analysis by specifically focusing on CpG methylation located in the LTR sequence of these elements. This is where DNA methylation has been shown to be targeted and to matter for retrotransposons repression. Changes may be greater.

We determined CpG methylation separately on LTRs and internal regions. However, the loss of CpG methylation at ETn LTRs was actually less pronounced than in the internal regions. We believe that the KRAB-ZFP binding prevents demethylation at the immediate binding sites and all ETn-targeting KRAB-ZFPs we identified bind the internal ETn region and not the LTRs.

As for IAP, we saw a substantial decrease (20% of CpG sites) in methylation in one type of IAP LTR (IAPLTR4) which is directly targeted by a Chr4-cl KRAB-ZFP (Gm13157). However, the other targeted IAP LTRs (e.g. by Gm21082) did not display any methylation loss family-wide, most likely due to redundancy of KRAB-ZFPs that bind IAP elements. Preliminary data we have collected indicates however that a small subset of IAP LTR-subtypes that are bound by Gm21082, lose methylation in Chr4 KOs.

5) The data graphed in Figure 4B do not seem to reflect the pedigree depicted in Figure 4A. Only 24 progenies could be counted for the KO/WT x KO/WT matings (12 WT/WT or WT/KO, and 12 KO/KO). In addition, some data points (such as the second from top, x=KO, y=~65) seem to originate from a different type of cross (heterozygote x homozygote). The overall conclusion (that KO produce on average more ETn insertions) could be quite different if a small number of individuals was misplaced in the graph, given the size of the samples. It is important to make sure which animals were accounted from the pedigree in 4A to produce the boxplot 4B.

The data in Figure 4B indeed comes from mice derived from three different mating (two KO/WT x KO/WT and one KO x KO/WT in the B6/129 F2 background). Not all animals from these matings were included in the analysis, only the ones where the DNA was purified from juvenile tail tissue. We did that to exclude a potential bias since tail samples showed generally more novel insertions than other tissues (Figure 4—figure supplement 3C).

We now show the data separated by matings as well as combined, and describe the data in the main text. This analysis revealed that in only one of the three matings, the difference in ETn insertions between KO/WT and KO reaches significance (p value < 0.05). We re-wrote the paragraph in the main text accordingly.